# Therapeutic Targeting of Collective Invasion in Ovarian Cancer

**DOI:** 10.3390/ijms20061466

**Published:** 2019-03-22

**Authors:** Laura Moffitt, Nazanin Karimnia, Andrew Stephens, Maree Bilandzic

**Affiliations:** 1Hudson Institute of Medical Research, Clayton VIC 3168, Australia; lrmof1@student.monash.edu (L.M.); nazanin.karimnia@monash.edu (N.K.); andrew.n.stephens@hudson.org.au (A.S.); 2Department of Molecular and Translational Sciences, Monash University, Clayton VIC 3800, Australia

**Keywords:** ovarian cancer, leader cells, metastasis, therapies, invasion

## Abstract

Ovarian cancer is the seventh most commonly diagnosed cancer amongst women and has the highest mortality rate of all gynaecological malignancies. It is a heterogeneous disease attributed to one of three cell types found within the reproductive milieu: epithelial, stromal, and germ cell. Each histotype differs in etiology, pathogenesis, molecular biology, risk factors, and prognosis. Furthermore, the origin of ovarian cancer remains unclear, with ovarian involvement secondary to the contribution of other gynaecological tissues. Despite these complexities, the disease is often treated as a single entity, resulting in minimal improvement to survival rates since the introduction of platinum-based chemotherapy over 30 years ago. Despite concerted research efforts, ovarian cancer remains one of the most difficult cancers to detect and treat, which is in part due to the unique mode of its dissemination. Ovarian cancers tend to invade locally to neighbouring tissues by direct extension from the primary tumour, and passively to pelvic and distal organs within the peritoneal fluid or ascites as multicellular spheroids. Once at their target tissue, ovarian cancers, like most epithelial cancers including colorectal, melanoma, and breast, tend to invade as a cohesive unit in a process termed collective invasion, driven by specialized cells termed “leader cells”. Emerging evidence implicates leader cells as essential drivers of collective invasion and metastasis, identifying collective invasion and leader cells as a viable target for the management of metastatic disease. However, the development of targeted therapies specifically against this process and this subset of cells is lacking. Here, we review our understanding of metastasis, collective invasion, and the role of leader cells in ovarian cancer. We will discuss emerging research into the development of novel therapies targeting collective invasion and the leader cell population.

## 1. Ovarian Cancer: A Unique Mode of Metastasis

Whilst the molecular mechanisms driving metastasis are often similar across different tumour types, in ovarian cancer, hematogenous intravasation/extravasation comes secondary to passive peritoneal dissemination. Indeed, even the most aggressive, high-grade ovarian cancers rarely metastasize beyond the peritoneum, and this remains a poorly understood characteristic of the disease [1,2,3,4].

Local invasion of ovarian cancer cells to neighbouring tissues occurs by direct extension from the primary tumour; whereas dissemination to distal sites within the peritoneum occurs by passive movement of ovarian cancer spheres within the peritoneal fluid or ascites [5]. In the latter route, ovarian cancer cells destined for exfoliation from the primary tumour acquire a unique expression profile, where both epithelial and mesenchymal markers are co-expressed. This “cadherin switch” involves the overexpression of transcription factors including ZEB1, TWIST, and Slug and Snail resulting in the upregulation of E-cadherin, activation of mesenchymal markers N-cadherin and Vimentin, and acquisition of an epithelial–mesenchymal transition (EMT)-like phenotype [6,7]. The remodelling of the ovarian epithelium is further dependent on integrin-mediated upregulation of matrix metalloproteinases (MMPs), which in turn facilitate the ectodomain shedding of E-cadherin, resulting in decreased cell–cell adhesion and the detachment of ovarian cancer cells from the primary tumour into the peritoneal cavity (Figure 1). Within the peritoneal cavity, ovarian cancer cells tend to form multicellular aggregates termed “spheroids” [8]. The presence of anchorage-independent spheroids complicates disease management and indicates a poor prognosis, as spheroids exhibit an increased propensity to survive chemotherapies and seed multiple distal metastases [9,10].

Whilst establishing secondary nodules, metastatic ovarian cancer cells interact with the single-cell layer of mesothelium lining the peritoneal cavity and organs, superficially attaching to and invading the underlying matrix [2,4,11]. In the period between apposition at the peritoneal lining and invasion of the underlying extracellular matrix (ECM), transcriptional “reprogramming” switches tumour cells from a proliferative to invasive physiology to facilitate degradation of the underlying matrix [12]. This process occurs universally in all ovarian cancer patients, the majority of whom are initially diagnosed with metastatic disease and persists in the >90% of patients who experience relapse following treatment. Spheroid adhesion to peritoneal surfaces is mediated directly through interactions between the cancer spheroid and receptors on the surface of the mesothelial layer. Decreased E-cadherin expression on the outer surface of the spheroid induces the expression of adhesion receptor molecules including CD44 and several integrins [13,14,15], priming spheroids for subsequent attachment to ECM proteins on the surface of the mesothelium [2,4,11,16]. Studies have shown that the interaction between spheroid expressed α5β1-integrin and mesothelial expressed fibronectin is essential for spheroid adhesion to the mesothelium [17,18]. Likewise, αvβ3-integrin was shown to be key to the proliferative and invasive behaviour of ovarian cancer cells [19]. In vitro inhibition of the α3, α6, and β1 integrin subunits in ovarian cancer spheres decreased invasiveness and collagen binding. Further, the inhibition of α2β1-integrin abolished the ability of ovarian cancer spheres to disaggregate on an artificial ECM [20,21]. 

Studies examining CD44 blockade demonstrated a reduction in the number of secondary tumours formed, but it was not sufficient to inhibit the mesothelial adhesion of ovarian cancer cells [15]. Cell adhesion molecule L1 (L1CAM) has also been shown to modulate the adhesion of ovarian cancer spheroids to the mesothelium by interacting with mesothelial neuropilin-1 receptors (NRP-1) [22]. An L1CAM specific antibody reduced murine peritoneal metastases [23] and prolonged survival when used in conjugation with radiotherapy in animal studies [24]. Further, the fractalkine receptor (CX_3_CR1) expressed by ovarian cancer cells has also been demonstrated to mediate ovarian cancer cell adhesion to mesothelium by interacting with its ligand CXCL1 present on the surface of mesothelial cells’ surface [22]. siRNA-mediated downregulation of CX_3_CR1 reduced the adhesion of SKOV3 cells to mesothelial LP9 cells by 50% [25]. Once ovarian tumour spheroids are attached to the mesothelium, they initiate infiltration and spread to surrounding tissue. 

## 2. Collective Invasion and Leader Cells

Multicellular clusters of ovarian cancer cells migrate in a directed and coordinated fashion in a process called collective invasion [26]. Three key characteristics defining collective invasion are: (i) the preservation of the physical connections and cell–cell junctions to orchestrate collective movement; (ii) shared cytoskeletal dynamics, allowing groups of cells to proceed as a single unit and develop multicellular polarity; and (iii) interactions with other cells and the ECM along the migratory path [26,27,28]. Collective invasion is a fundamental property of many metastatic tumours in human cancers, particularly epithelial tumours [29,30] including pancreatic cancer [31], colon cancer [32], sebaceous cancer [33], melanoma [34], breast cancer [35,36,37], and lung cancer [38].

Despite the applicability of the basic principles governing single-cell migration to collective migration, the molecular events controlling collective invasion are far more intricate and not all cells within the cluster are invasion-competent [26]. Collective invasion is driven by “leader cells”, a functionally distinct sub-population of cancer cells that direct migration, promote changes in cellular contractility, and lead the trailing “follower” cells [39]. In breast and bladder cancers, mesenchymal-like leader cells maintain distinct cellular polarity, form protrusive filopodia, and respond dynamically to environmental cues [28,37,40]. The follower population retains expression of adhesion junctional proteins and maintain a packed morphology [36,41].

During collective invasion (illustrated in Figure 2) cytoskeletal polarisation establishes a front–rear axis within the cluster. Leader cells at the front axis undergo supra-cellular cytoskeletal organization by rearranging their actin filaments, facilitating membrane protrusion and the formation of invadopodia. This process requires activation of phosphoinositide 3-kinase (PI3K) [42] and GTPase proteins, including cell division cycle 42 (Cdc24) and Ras-related C3 botulinum toxin substrate (Rac), to rearrange the cytoskeleton and induce actin expression. Concurrently, Rho signalling at the posterior side mediates actomyosin contraction, generating the force required for cellular movement [28,43]. The leader cells then enable the penetration of the basement membrane comprised of collagen I and IV, laminin, and fibronectin [44] through the expression of proteolytic enzymes including MMPs and other serine proteinases [45].

In addition to proteolytic enzymes, cancer associated fibroblasts (CAFs) are abundantly present within the tumour microenvironment (TME) and play a crucial role in mediating collective invasion. Further, CAFs induce changes including physically remodelling the TME to lay “tracks” for invading cancer cells [46]. Cancer associated fibroblasts further drive collective invasion, by mediating the heterophilic adhesion between membrane E-cadherin on the tumour cell and N-cadherin on the CAF [47]. These changes within the TME further ensure the epithelial phenotype of invading cells is retained [48].

## 3. Leader Cells and Progenitor-Like Properties

A unique feature of leader cells is their high degree of transcriptional plasticity, making them distinct from cells undergoing EMT and the follower cell population alike. In breast cancer, leader cells co-express multiple basal epithelial markers such as KRT14, KRT5, P63, and P-cadherin, in conjunction with markers of the luminal epithelium including KRT8, KRT18, and E-cadherin, marking them as a progenitor-like population [37]. Leader cells may also secrete immune effector molecules, influencing lymphocyte differentiation and polarizing local immune responses towards a suppressive phenotype [49]. Several studies have now demonstrated the importance of leader cells in the progression of epithelial-tumour types including breast, salivary, bladder, prostate, and lung cancers [36,38,49,50,51].

Several lines of evidence suggest that KRT14+ cells possess similar features to cancer stem cells (CSCs). In a bladder regeneration mouse model, KRT14+ cells exhibited increased clonogenicity and gave rise to multiple differentiated progeny; in fact, this KRT14+ cell population was an absolute requirement for the re-establishment of epithelial cell layers following damage [49]. Similarly, KRT14+ cell “stemness” was demonstrated in vivo using breast cancer mouse models [36]. In particular, KRT14+ cells were highly enriched in disseminated micro-metastases (accounting for over 50% of cells present); over time, however, the KRT14+ cell population reverted to baseline levels, in keeping with their transcriptional plasticity [36]. Other studies have also demonstrated that leader cells are enriched in response to chemotherapy and promote the acquisition of chemo-resistance over time, characteristics reflective of CSC status [37,52,53]. Together, these studies highlight the requirement of leader cells for metastasis and their potential roles as CSCs in epithelial tumours [54]. Our own studies have shown KRT14 expression is confined to the leading edge of ovarian cancer cells in both 2D and 3D format with KRT14 ablation rendering ovarian cancer cells invasion incompetent.

## 4. Targeting Leader Cells as a Novel Approach to Ovarian Cancer Management

Current first-line treatment methods for high grade serous ovarian cancer involve surgical debulking and adjuvant chemotherapy using platinum and taxane drugs. Unfortunately, remission is generally short-lived, and recurrence occurs in at least 90% of patients [55]. Several emerging therapies and combination strategies have shown initial promise, but as of yet, none have achieved long-term disease regression in the broader context of disease. Ultimately, repeated rounds of chemotherapies result in the emergence of chemo-resistant disease for the vast majority of patients, limiting further available treatment options.

Whilst the majority of traditional anti-cancer drugs target the proliferative behaviour of tumour cells, it is the invasive propensity of malignant cells that leads to metastasis and ultimately accounts for overall morbidity and mortality [56]. As the drivers of solid cancer migration and metastasis, inhibition of the leader cell component represents an attractive and potentially promising new approach for cancer treatment. In particular, therapies designed to eliminate leader cells in tumours are likely pivotal to achieve sustained remission for patients with ovarian cancers. However, complete definition of the transcriptional, epigenetic, and proteomic signatures of leader cells—in a variety of solid cancer types—is required before specifically targeted molecular therapies become available.

Below, we discuss existing, emergent, and potential therapeutic approaches that may be effective for targeted leader cell depletion.

## 5. Current Standard-of-Care in Ovarian Cancer Therapy

Amongst a plethora of ongoing clinical trials, immunotherapies (including checkpoint inhibitors and antibodies against growth factors or signalling molecules) and PARP inhibitors have proven the most widely examined for their potential to manage an otherwise terminal disease. Immune checkpoint inhibitors (e.g., Keytruda^TM^ and Avelumab^TM^) and immune modulators that have been successful in other cancer types have shown only limited efficacy in ovarian cancer trials [57,58]. Likewise, epidermal growth factor and folate receptor targeting therapies have revealed disappointing results [59,60]. Antiangiogenic therapies have only shown a marginal increase to progression-free survival (PFS) (2–6 months) outweighed by both the high cost involved with drug administration and the toxic side-effects experienced by patients [61].

Achieving greater success are the PARP inhibitors, indicated for the treatment of ovarian cancer patients carrying BRCA mutations [62]. In ovarian cancer patients with hereditary BRCA mutations, the PARP inhibitor Olaparib^TM^ significantly increased PFS from 5.5 to 30.2 months [63]. Similarly, the PARP inhibitor Niraparib^TM^ improved PFS in all ovarian cancer patients; with the highest efficacy observed in BRCA mutation carriers (increased PFS by 15.5 months compared to the placebo [64]). Each of these inhibitors are associated with adverse haematological effects, readily managed by dose modification [64]. The impact of these drugs in non-BRCA mutated, chemo-resistant patients is the subject of ongoing clinical trials. The PARP inhibitors are now the standard of care for patients with recurrent, BRCA-mutated ovarian cancer; however, they bear costs that are up to eight times that of platinum-based care.

## 6. Targeting Leader Cell-Directed Processes: Collective Invasion and the Invasive Front

Metastatic disease is commonly associated with resistance to conventional treatments such as chemotherapy [65]. Fundamental to metastasis are the molecular and morphological changes associated with EMT during invasion. The transient nature of EMT and difficulties in comprehensively defining this state means that the inhibition of EMT-related processes is challenging; however, and few druggable targets have been identified [66]. By contrast, collective invasion and the processes of intra- and extravasation are a common feature of many solid cancer types including ovarian [67,68]; and increasing evidence suggests that leader cells are pivotal drivers of these processes. Directly targeting leader cells, and the processes they regulate may thus serve as a promising approach to combat metastatic disease.

The complex nature of metastasis suggests that multiple therapeutic targets should exist for directed interventions. Indeed, ongoing research has targeted two key areas: disruption of collective invasion and the impairment of attachment and invasion-related processes at the spheroid–mesothelial interface. Clear molecular definition of the leader cell phenotype is required, however, before specifically directed molecular therapies aimed at ablation of these cells in established tumours can be developed.

## 7. Strategies to Inhibit Collective Invasion in Ovarian Cancers

### 7.1. Cytoskeletal Stability 

Several studies have examined the potential of a “migrastatic” approach, using drugs broadly targeting cytoskeletal stability (e.g., actin polymerization or stabilization), cellular contractility, and ion transport to prevent the formation of invadopodia, and thus inhibit metastasis [69]. Whilst promising in vitro for the inhibition of lung, melanoma [70], and prostate cancer [71] outgrowth, the failure of anti-actin drugs to discriminate between malignant and normal cells in vivo presents an unfavourable toxicity profile [70,71]. A second class of inhibitors, targeting actin-binding proteins tropomyosin and myosin to inhibit actin assembly and function, have also been proposed as a less toxic, more suitable approach [72]. For example, the anti-tropomyosin compound TR-100 specifically disrupts the cytoskeleton of tumour cells and has been shown to effectively reduce tumour growth in melanoma and neuroblastoma mouse models [73]. Collective invasion may also be targeted by the myosin inhibitor Blebbistatin that works to decrease myosin activity in non-muscle cells. This compound has been shown to be effective at inhibiting invasion in a wide range of cancer cell lines; however, there are no published in vivo data available and its application has not been demonstrated for ovarian cancer [74].

### 7.2. Rho Kinase Inhibition 

Targeting the specific kinases and phosphatases that regulate phosphorylation cascades involved in actin polymerisation are also a potentially useful approach [75,76]. Rho GTPases are important regulators of cytoskeletal dynamics and play a significant role in cancer cell migration and invasion [77]. Highly localised RhoA activity is observed at the leading tip of invadopodia emerging from kidney epithelial cells, associated with actin filament activation during invasion [78]. Whilst not yet demonstrated in ovarian cancer, RhoA is overexpressed in metastatic omental ovarian cancer deposits [79]. Moreover, RhoA overexpression increases ovarian cancer cell invasiveness in vitro, and nude mice implanted with RhoA overexpressing cells developed a significantly greater number of disseminated tumours [80]. Accordingly, knockdown of RhoA decreased migration and invasion in vitro and reduced ascites accumulation and peritoneal dissemination in nude mice [81].

Interestingly, a recent study reported decreased RhoA expression in lung cancer leader cells following silencing of KRT14 [82]. These cells exhibited decreased invasion and migration in vitro, strongly suggesting a key role for RhoA/Rho-associated kinase (ROCK) signalling in leader cells [82]. Accordingly, targeting of RhoA signalling by the HMG-CoA reductase inhibitor Lovastatin re-sensitized chemo-resistant ovarian cancer cell lines to doxorubicin in vitro [83]. The combination of Lovastatin with chemotherapy has yet to be tested in clinical trials.

ROCK, a downstream effector of RhoA signalling, has also been suggested as a therapeutic target for the management of metastatic disease. Following activation by RhoA, ROCK I/II phosphorylates substrates including the myosin phosphatase (MYPT1) and myosin regulatory light chain (MLC) to promote cytoskeletal rearrangement and cellular contractility [84]. The ROCK inhibitor Fasudil (HA-1077) [85] has shown promising results in vitro and in vivo in several cancer types including brain, lung, liver, and ovarian cancers [86,87,88,89]. Tumour priming with Fasudil in pancreatic cancer mouse models and patient-derived xenografts improved response to chemotherapy at secondary sites and reduced metastatic spread [90].

In ovarian cancers, Fasudil attenuated lysophosphatidic acid (LPA)-induced ovarian cancer cell migration and invasiveness and reduced the intraperitoneal spread of cancer cells in a mouse xenograft model [91]. Similarly, the ROCK inhibitor Y27632 [92] also decreased LPA-induced invasiveness of ovarian cancer lines Caov3 and PA-1 [93]. Fasudil also enhanced sensitivity to cisplatin in A2780 ovarian cancer cells in vitro, suggesting its potential synergy with chemotherapy [94]. Whilst Fasudil has not progressed to human trials for metastatic disease, the novel ROCK I/II inhibitor AT13148 [95] is currently in phase I clinical trials for patients with advanced breast, prostate, and ovarian tumours (NCT01585701). AT13148 induced a wide range of side-effects impacting on the cardiovascular system including smooth muscle contractility, tachycardia, and high-blood pressure, which were alleviated by modifying the administered dose [96]. The effects of Rho/ROCK inhibition are likely to be cancer-type-specific; however, for example, by contrast to results in ovarian cancer cells, Y27632 increased the invasive potential of human glioma U87 and U251 cells [97] and gastric carcinoma OCUM-2MD3 cells [98]. Indeed, ROCK activation is not always oncogenic, and the specific role of ROCK is cell-type and tumour-microenvironment dependent [99].

### 7.3. Other Kinase Targets

Alongside established roles in proliferation and angiogenesis, Src-kinase is now gaining interest as a mediator of cellular motility and the formation of invadopodia. Src is overexpressed in several solid cancers including breast, colon, and ovarian cancer [100,101], and is essential for breast cancer invadopodia formation via downstream regulators such as cortactin and Tsk5 [102]. Cortactin is localised in breast cancer cell invadopodia, where it regulates actin stabilisation and the recruitment of ECM proteases to the invasive interface [103]. In vitro studies using the Src inhibitors dasatinib and saracatinib showed high efficacy in preventing metastasis in several cancer models including pancreatic, prostate, and ovarian cancers [104]. Unfortunately, these inhibitors did not reach their anticipated heights in clinical trials where phase II trials of Dasatinib in women with recurrent epithelial ovarian cancer or peritoneal carcinoma (NCT00671788) showed no significant increase to PFS [105].

Other multi-kinase inhibitors such as cabozantinib and sorafenib are currently being evaluated in clinical trials for the treatment of metastatic epithelial ovarian cancer; however, most have been met with limited success and exhibit significant toxicity (Table 1) [106,107,108,109,110,111]. The potential off-target effects associated with inhibition of cell contractility present a major challenge for the development of these “anti-metastatic” drugs, where effects on normal physiological processes including wound healing, cell division, and the immune cells must be minimized. Despite the development of second-generation kinase inhibitors designed to increase specificity and limit the off-target effects on normal tissue, their efficacy in vivo has yet to be demonstrated [112].

## 8. Strategies to Disrupt Attachment and Invasion at the Invasive Interface

As key regulators of the metastatic cascade, leader cells define the interactions that occur between the invasive cancer deposit, ECM proteins at the target site, and the underlying healthy tissue sub-stratum tissue. During invasion, emerging cancer invadopodia comprised of leader cells must attach and degrade extracellular components including collagen IV, laminin, and proteoglycans to successfully overcome the basement membrane barrier and infiltrate peripheral tissue [125]. As previously mentioned, integrin-mediated interactions with ECM proteins on the mesothelial interface are essential for the adhesion of ovarian cancer spheroids at the site of invasion. In a xenograft mouse model of ovarian cancer, the anti-α5β1-integrin antibody Volociximab reduced tumour burden and was well tolerated in a phase I clinical trial [126]. However, in a phase II clinical trial in women with platinum resistance and advanced epithelial ovarian cancer, volociximab was not significantly effective at preventing disease progression and was associated with adverse events (NCT00516841) [120]. Other antibodies have also been developed to target the αv-integrin family. The pan-anti-αv-integrin antibody intetumumab was shown to inhibit adhesion, migration, and invasion in breast cancer cells in vitro and reduced tumour growth and metastasis in mouse models [127]. Intetumumab treatment in stage IV melanoma patients did not significantly improve overall survival [128]. Similarly, etaracizumab (anti-αvβ3-integrin) only demonstrated a marginal increase to PFS in a phase II trial of metastatic melanoma [129]. Despite their promise in preclinical studies, anti-integrin antibody therapies have failed to demonstrate clinically meaningful improvements to PFS compared to standard treatments thus far, and have not progressed to trials in ovarian cancer patients [130].

In addition, MMPs are actively involved in the proteolytic degradation of the ECM and are commonly overexpressed in invasive cancers [131]. A key mediator of collective invasion is MT1-MMP (MMP-14), which accumulates at the invasive front of tumours and is upregulated on the surface of breast cancer invadopodia [102,132]. MT1-MMP directly degrades ECM components in addition to activating other MMPs at the tumour–stromal interface [133]. Most MMP inhibitors have limited specificity resulting in extensive off-target effects, and this class of inhibitors have been largely unsuccessful in clinical trials [45]. The monoclonal antibody DX-2400 specifically inhibits MT1-MMP and has shown promise by blocking invasion and migration in vitro. Further, in vivo DX-2400 delayed tumour growth and metastasis alone and in combination with paclitaxel in mouse breast cancer xenograft models [134]. However, this potent MT1-MMP inhibitor has yet to progress to clinical trials. 

An alternative approach to directly targeting ECM proteases such as the MMPs may be to inhibit proteins involved in their intracellular trafficking or regulators of their expression. For example, multiple studies have proposed that the molecular mechanisms of inhibitors of RhoA and ROCK (described above) involve the downregulation of MMP-2 and MMP-9 which are present at the invasive front of many cancers [88,135]. Furthermore, our own preliminary studies in ovarian cancer have indicated that MMP-10 and MMP-13 may interact with the serine protease DPP4 at the cell surface, for which there are clinically available inhibitors.

## 9. Molecular Targets in Leader Cells

Although the targets described above are involved in the general metastatic pathway, their specific relationship to the leader cell phenotype is not completely understood. In particular, the molecular genetic phenotype of leader cells may differ between cancer types, disease location, grade or stage; thus, the identification of exclusive, cell-type-specific targets represents one of the greatest barriers for the development of directed, clinically relevant therapies.

Despite these limitations, however, new targets are emerging. Yamaguchi et al. [136] recently identified active upregulation of Rac, PI3K, and integrin β1 as specific markers of epithelial kidney leader cells that contribute to collective cell migration. Pharmacological inhibition of each of these targets individually was effective in disrupting collective cell migration. Specifically, the authors were able to demonstrate that the migration of leader cells was dependent on Rac1 activity and was likely via a downstream feedback loop involving the independent actions of PI3K and integrin β1 [136]. Rac1 and associated signalling molecules are overexpressed in epithelial ovarian cancer.

High-throughput screening and structural homology approaches identified a small molecule inhibitor, R-Ketorolac, which specifically targets Rac1 and the related protein Cdc42. In ovarian cancer cell lines, R-Ketorolac was found to decrease cell adhesion, migration, and invasion and effectively prevented Cdc42-dependent invadopodia formation [137]. In a retrospective “Phase 0” trial, ovarian cancer patients receiving ketorolac for post-operative analgesia had an increased overall survival probability compared to the placebo-treated control group, where 18% of Ketorolac-treated patients succumbed to the disease after 5 years, compared to 43% of non-treated patients [123]. The specificity and efficacy of post-operative Ketorolac in a small cohort of ovarian cancer patients undergoing cytoreductive surgery is the subject of an ongoing clinical trial in the recruitment phase (NCT02470299).

Metastatic disease is the most significant challenge to the management of all types of cancer and accounts for more than 90% of cancer-related suffering and death. The vast majority of ovarian cancer patients are diagnosed with late-stage metastatic disease, and current treatment methods are not effective as most patients experience relapse. We anticipate future studies defining this unique subset of leader cells will facilitate the development of effective targeted therapeutics. Leader cell targeting serves as an exciting emerging area of research offering highly novel approaches to promote tumour regression and increase management options available for late-stage disease.

## Figures and Tables

**Figure 1 ijms-20-01466-f001:**
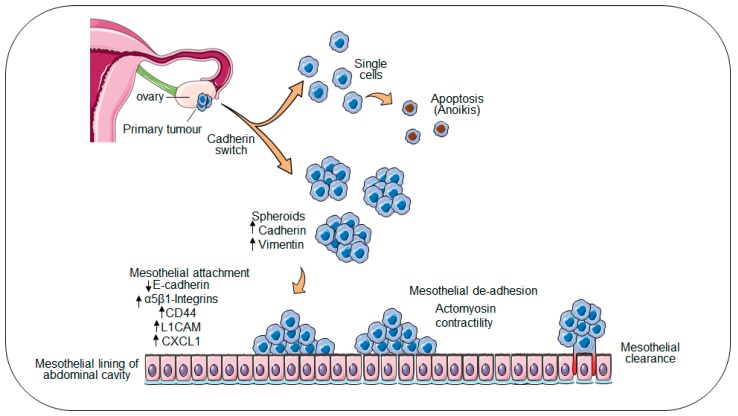
Metastasis model in ovarian cancer. A schematic model of ovarian cancer progression and dissemination. Ovarian cancer cells in the primary tumour acquire a unique expression profile and are exfoliated from the primary tumour site into the ascites. Ovarian cancer cells which have shed form multicellular aggregates are termed spheroids.erin. Spheres are carried passively within the peritoneum by the peritoneal fluid or ascites where they seed multiple distal metastasis by attaching to and clearing the mesothelial lining.

**Figure 2 ijms-20-01466-f002:**
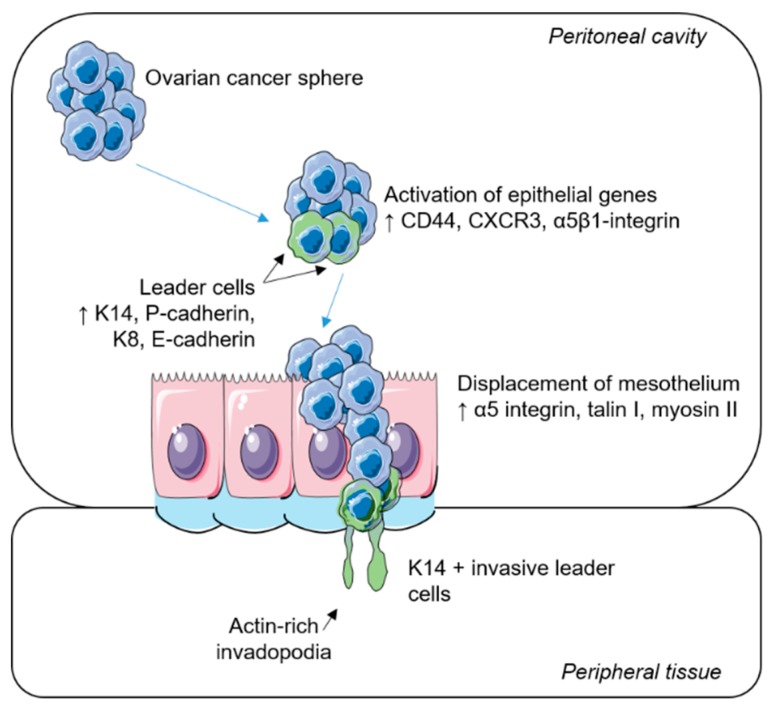
Collective invasion in epithelial ovarian cancer. A schematic representation of the metastatic spread of ovarian tumour cells. Ovarian cancer spheres diffuse throughout the peritoneal cavity. Upon their attachment to the mesothelial layer, epithelial genes are activated. Specialized leader cells transiently express basal epithelial and luminal epithelium markers and displace the target mesothelium via the formation of actin-rich invadopodia, where trailing cells follow to colonize surrounding tissues.

**Table 1 ijms-20-01466-t001:** Selected targeted drugs in clinical trials for the treatment of metastatic ovarian cancer.

Drug	Target	Clinical Trial ID	Phase	Outcome Measures	Current Status	Refs
Lovastatin	RhoA	NCT00585052	II	Tumour response rate in combination with paclitaxel for patients with relapsed ovarain cancer	Terminated due to slow accrual.	[113]
AT13148	Multi-AGC kinase	NCT01585701	I	Determine dosing and adverse events. Evaluate any response in patients with advanced-stage solid tumours including prostate, breast, and ovarian.	Completed. Preliminary data indicate tolerable on target effects.	[96,114]
Dasatinib	Src kinase	NCT00671788	II	Progression Free Survivial at 6 months and tumour response in persistent or recurrent epithelial ovarian cancer using dasatinib as a monotherapy	Completed. dasatinib has minimal activity as a single agent in ovarian cancer (PFS 2.1 months).	[105,115]
Cabozantinib	Multi-kinases	NCT00940225	II	Evaluate overall response rate and PFS in patients with advanced malignancies including melanoma, breast and ovarian cancer	Completed. Ovarian cancer patients showed the highest overall response rate (21.7%) and disease control rate was 50%. Platinum-sensitive patients achieved a longer PFS (6.9 months) than platinum-resistant patients (2.8 months).	[106,108,116]
		NCT01716715	II	Compare PFS in patients with persistent or recurrent ovarian cancer patients receiving cabozantinib or paclitaxel	Ongoing.	[117]
Sorafenib		NCT00093626	II	Assess adverse events and PFS time in patients with persistent or recurrent ovarian cancer	Completed. Significant toxicity as a monotherapy with modest anti-tumour effect (PFS 2.1 months).	[109,118]
		NCT00526799	I/II	Tolerability (Phase I) and response rate (Phase II) to treatment with sorafenib in combination with topotecan in patients with platinum-resistant or refractory-recurrent ovarian cancer	Terminated. Significant toxicity caused by sub-optimal doses of combination therapy associated with minimal clinical efficacy.	[110,119]
		NCT00390611	II	PFS over 2 years in patients with late-stage ovarian cancer receiving sorafenib in first-line treatment	Completed. Combination paclitaxel/carboplatin or paclitaxel/carboplatin/sorafenib had similar response rates and PFS (15.4 vs 16.3 months). The addition of sorafenib in first-line treatment caused increased toxicity.	[111,119]
Volociximab	α5β1-integrin	NCT00516841	II	Evaluate efficacy of volociximab monotherapy by objective response rate and tumour response in patients with platinum resistant EOC	Terminated due to insufficient clinical activity. Volociximab was well tolerated; however, there were no complete or partial responses.	[120,121]
Ketorolac	Rac1/Cdc42	NCT01670799	0 (Pilot)	Determine measurable R- and S-ketorolac in post-operative treated patients following cytoreductive ovarian cancer surgery	Ongoing.	[122]
		NCT02470299	I	Confirmation of drug specificity. Evaluation of overall survival and PFS in post-operative IV ketorolac treated ovarian cancer patients	Recruiting/ongoing. Preliminary data shows specific Rac1 and Cdc42 inhibition and potential prolonged survival in women receiving ketorolac.	[123,124]

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
