# Peer review of "Therapeutic Targeting of Collective Invasion in Ovarian Cancer"

_ijms, 2019, doi:10.3390/ijms20061466_

Round 1

Reviewer 1 Report

This review manuscript by Moffitt et al. elegantly discussed the mechanisms and clinical implications of collective behavior of ovarian cancer cells, the significance of which has been appreciated by many investigators in the field of cancer research. As the authors of this manuscript are aware of, cancer cells that undergo collective migration share the same mechanisms as those undergo single cell migration, and there have been few studies that have dealt with mechanisms that are specific to collective cell migration or invasion. Ovarian cancer-specific mechanisms of collective invasion have also not been unraveled, which warrants further studies in this field. The authors pointed out in this review article that cytoskeletal stability, Rho kinase inhibition, and Src signaling are critical for collective invasion of cancer cells, all of which I believe are useful for the readers. The authors also discussed the importance of leader cells in cancer cell cohorts or groups, suggesting the existence of some mechanisms that regulate gene expression profile of leading cells that is distinct from that of following cells. Obviously, the expression of some keratin subsets seems to be critical to specify the function of the leading cells.

     Overall, I appreciate that this review article is a great one that covers many aspects of collective invasion, although most of the contents are not specific to ovarian cancer. Some points needs to be addressed before publication.

1) Recent observations from several groups showed the importance of the tumor microenvironment (TME) in determining the behavior of cancer cell groups. Specifically, cancer-associated fibroblast (CAFs) or tumor-associated fibroblasts (TAFs) have critical functions to remodel the extracellular matrix and make a path or tunnel for migrating cancer cell groups. Importantly, direct physical interactions between CAFs/TAFs and cancer cells through E-cadherin/N-cadherin interaction have also been revealed. In this regard, the authors need to add a section to describe the contribution of CAFs/TAFs in determining the behavior of collective cancer cell groups by citing milestone papers including for example studies by Erik Sahai’s group (Gaggioli et al., Nat Cell Biol 9, 1392-1400, 2007), Xavier Trepat’s group (Labernadie et al., Nat Cell Biol 19, 224-237, 2017), and Genichiro Ishii’s group (Neri et al., Int J Cancer 137, 784-796, 2015).

2) As mentioned in the manuscript, there have been few studies that described the mechanisms that are specific to collective migration of cells but not broadly relevant to single cell migration. In this regard, the authors could discuss the significance of “supracellular” cytoskeletal organization, which regulates not only cell-cell adhesion but also coordinated remodeling of the actin cytoskeleton across cells that constitute the cell group. The importance of “supracellular” cytoskeletal organization has been described by pioneers in this field including Peter Friedl and his colleagues (as cited by the authors in ref. 31 and 47). In addition, specific mechanism of collective migration has also been beginning to be shown very recently. A recent paper from a Japan group showed that an actin-cytoskeletal binding protein girdin, which is a regulator of neuroblast collective migration in neural development, is also crucial for supracellular cytoskeletal organization and collective cancer cell migration (Cancer science, 109, 3643-3656, 2018). I am not following all papers in this field, but the regulatory mechanisms of “supracellular” cytoskeletal organization is essential when we talk about the mechanism of collective cancer cell migration.

3) As the authors mentioned, the expression of keratin 14 in the leading cells of cancer cell groups is intriguing, which was pioneered by Andrew Ewald group (Cell, 155, 1639-1651, 2013). Are there any evidence showing that keratin 14 expression is also found in the leading cells of ovarian cancer? This information could be appreciated by the readers given the title of this review manuscript.

Author Response

We sincerely thank you for taking the time to provide such excellent feedback.  We found your review to have so many insightful comments it would be amazing to pick your brain!  

We have attempted to incorporate your changes and suggestions. The changes can be found in red in the main manuscript file.

1.  We have included an additional paragraph detailing the role of CAFs in remodelling the TME for collective invasion (paragraph following the supracellular organization  (page 3). 

2. We have made mention of the supracellular organization process also  paragraph (following figure 2 page 3). 

 3. we have referred to our own unpublished studies demonstrating the localization of Krt14 to the leading edge of ovarian cancer cells (page 4). 

If there are any further details required please do not hesitate to contact me.

Reviewer 2 Report

Summary

This is a review on ovarian cancer metastasis, including metastasis via spheroids and the possibility of targeting leader cells as a therapeutic target in ovarian cancer.

General Comments:

The review is well written and easy to understand.

The title focuses very specifically on the idea of targeting leader cells in ovarian cancer treatment. However, discussion on targeting leader cells is a relatively small part of the review (probably because of a lack of published literature on that specific area). Perhaps a title (and abstract) more focused on a slightly broader topic such as collective invasion of ovarian cancer (which mentions the possibility targeting leader cells) would be better describe the review.

The section on “Current standard-of-care in ovarian cancer therapy” should come earlier in the review, perhaps before the “Leader cells and progenitor-like properties” In its current location it breaks the train of thought on leader cells in ovarian cancer metastasis.

Specific Comments:

P6L4: should be “Highly localized RhoA”

P8L9: “signalling” should be “signaling”

Author Response

We thank you for taking the time to provide such great feedback.

We have attempted to incorporate all changes and suggestions. The changes can be found in red in the main manuscript file.

1.We totally agree with your comment “Perhaps a title (and abstract) more focused on a slightly broader topic such as collective invasion of ovarian cancer (which mentions the possibility targeting leader cells) would be better describe the review.” Accordingly we have changed the title and abstract to shift the focus to targeting collective invasion and have added additional text in the abstract to reflect this. 

2. We have decided to leave the standard of care where it is, we do feel shifting it to before the leader cell information takes a bit of the focus off the main part of the review. 

 3. We have made changes to our grammatical errors  and have changed the spelling for all instances of the work signaling.

Reviewer 3 Report

The paper by Moffitt and colleagues is well structured and organized. The subject matter is  of  interest for the scientific community and it  provides a good synthesis of the  literature of the subject matter. Thus, in the present form it can be accepted for pubblication.

Author Response

we thank you for taking the time to read and review our manuscript.